

# Dissociating predictability, plausibility and possibility of sentence continuations in reading: evidence from late-positivity ERPs

Laura Quante[1,2], Jens Bölte[1,2] and Pienie Zwitserlood[1,2]

[1] Department of Psychology, University of Münster, Münster, Germany
[2] Otto-Creutzfeldt-Center for Cognitive and Behavioral Neuroscience, University of Münster, Münster, Germany

## ABSTRACT

Late positive event-related potential (ERP) components occurring after the N400, traditionally linked to reanalysis due to syntactic incongruence, are increasingly considered to also reflect reanalysis and repair due to semantic difficulty. Semantic problems can have different origins, such as a mismatch of specific predictions based on the context, low plausibility, or even semantic impossibility of a word in the given context. *DeLong, Quante & Kutas (2014)* provided the first direct evidence for topographically different late positivities for prediction mismatch (left frontal late positivity for plausible but unexpected words) and plausibility violation (posterior-parietal late positivity for implausible, incongruent words). The aim of the current study is twofold: (1) to replicate this dissociation of ERP effects for plausibility violations and prediction mismatch in a different language, and (2) to test an additional contrast within implausible words, comparing impossible and possible sentence continuations. Our results replicate *DeLong, Quante & Kutas (2014)* with different materials in a different language, showing graded effects for predictability and plausibility at the level of the N400, a dissociation of plausible and implausible, anomalous continuations in posterior late positivities and an effect of prediction mismatch on late positivities at left-frontal sites. In addition, we found some evidence for a dissociation, at these left-frontal sites, between implausible words that were fully incompatible with the preceding discourse and those for which an interpretation is possible.

# INTRODUCTION

The study of effects of context on language processing has a long tradition in psycholinguistics, as modular (cf. *Forster, 1981*) and interactive (cf. *McClelland & Rumelhart, 1981*; *Marslen-Wilson, 1987*) theories of word recognition drastically differed with respect to the role allotted to information stemming from sources other than the word itself. Proof for an impact of contextual, top-down information on word recognition was already provided more than 30 years ago, with priming paradigms and reaction time data (cf. *Swinney, 1979*; *Schwanenflugel & Shoben, 1985*). The advent of event-related potentials (ERPs) again fired the debate, because they allow insights into

Corresponding author
Laura Quante,
l.quante@uni-muenster.de

the time-course of word recognition, which is difficult to come by with reaction times (but see *Zwitserlood, 1989*). Ever since, a wealth of studies has shown that contextual information, when constraining enough, has an early impact on lexical processing—even to the extent that upcoming words are anticipated (*Kutas & Federmeier, 2000*; *Van Berkum et al., 2003*; *DeLong, Urbach & Kutas, 2005*; *Van Berkum et al., 2005*). It is thus not surprising that terminology has changed, and "anticipation" and "prediction" are now used to refer to the impact, on lexical processing, of knowledge from sources other than the current input (cf. *Van Petten & Luka, 2012*; *Huettig & Janse, 2016*; *Kuperberg & Jaeger, 2016*). Whereas most researchers agree that (features of) upcoming words are predicted under certain circumstances, it remains unresolved which factors promote (or prevent) predictive processing, and what information about words (e.g., semantics, word forms) is predicted (see *Ito et al., 2016*; *Kuperberg & Jaeger, 2016*).

To study effects of semantic context, expectation and prediction in language comprehension, a particular ERP component, the N400 (*Kutas & Hillyard, 1980*), has been used extensively. The N400 is a negative-going wave peaking around 400 ms after stimulus onset, which is related to semantic processing (for a review, see *Kutas & Federmeier, 2011*). For example, its amplitude is negatively correlated to a word's cloze probability (proportion of respondents who completed a given context with this particular word), a measure of semantic expectancy. Words with strong contextual support show a decrease in N400 amplitude relative to words that are less predictable or do not fit the context (*Kutas & Federmeier, 2011*). There is also evidence for ERP effects as a function of predictability in time windows preceding the N400 (*Van Berkum et al., 2003*; *Dikker & Pylkkänen, 2011*; *Lau, Holcomb & Kuperberg, 2013*; *Brothers, Swaab & Traxler, 2015*; see *Kuperberg & Jaeger, 2016* for an overview). However, evidence for the actual pre-activation or anticipation of upcoming words, assessed before any of their input becomes available, is less abundant (but see *DeLong, Urbach & Kutas, 2005*; *Van Berkum et al., 2005*; *Szewczyk & Schriefers, 2013*; *Ito et al., 2016*).

Our study uses ERPs and does not focus on prediction or expectation per se, but on the consequences of prediction or expectation mismatch, and, more generally speaking, of contextual mismatch.[1] *Van Petten & Luka (2012)* proposed that if listeners and readers predict upcoming words, the Electroencephalographic (EEG) signal should reflect not only benefits of a confirmed prediction (visible as attenuation of the N400) but also costs of a disconfirmed prediction. In their review article, they assessed studies that compared congruent sentence completions with semantically anomalous completions, and often observed a late positivity, about 600–900 ms after critical-word onset, with a mainly parietal scalp topography. In addition, an anterior positivity was sometimes observed when ERPs for unexpected but semantically congruent sentence completions were compared to predictable, expected completions. It should be noted, however, that the 60+ studies included showed a great variability in the post-N400 time window.

It thus seems that unexpected continuations that allow construction of a possible overall sentence meaning differ from anomalous completions. Interestingly, studies that manipulate semantic expectancy have predominantly used anomalous or unexpected plausible completions, but rarely both. This motivated *DeLong, Quante & Kutas (2014)*

[1] We use "predictable" and "expected" interchangeably to characterize continuations that are highly expected given the preceding discourse, with predictability assessed by means of a cloze procedure.

to contrast different levels of plausibility within the same study, to determine how predictability and plausibility each contribute to word recognition. As completions of highly constraining sentence pairs (*For the snowman's eyes, the kids used two pieces of coal. For his nose, they used . . .*), DeLong et al. compared ERPs to highly predicable, expected (*a carrot*), unexpected but somewhat plausible (*a banana*) and unexpected, implausible, anomalous (*a groan*) words. The unexpected but plausible continuations should induce costs of disconfirmed prediction, combined with effort to integrate the unexpected noun—signaled by frontal late positivity. This does not hold for anomalous continuations that cannot be integrated with the current context. DeLong et al. observed a posterior late positivity to anomalous completions, and an anterior late positivity to unexpected but plausible completions, thus confirming *Van Petten & Luka's (2012)* conjecture. Corroboration for a particular function of the frontal late positivity, also labeled frontal PNP (post-N400 positivity), in prediction-related revision was recently provided by Swaab et al. (*Boudewyn, Long & Swaab, 2015*; *Brothers, Swaab & Traxler, 2015*).

Predictability thus seems to influence early stages of processing, whereas plausibility seems to affect late stages of processing, which is corroborated by eye-tracking studies (*Staub, 2015*, for an overview). Interestingly, *Rayner et al. (2004)* and *Warren & McConnell (2007)* further distinguished between plausibility and possibility, by comparing words that result in implausible but possible meaning for the full sentence, to words that induce an impossible overall sentence meaning, because they violate selection restrictions (e.g., "inflate a carrot"). In both studies, effects of words leading to either impossible or implausible sentence meaning were dissociable in eye-movement measures.

Processing differences between implausible and impossible sentence overall meaning are also visible in EEG data. For example, *Paczynski & Kuperberg (2012)* showed that selection-restriction violations evoked a posterior positivity between 700 and 900 ms after critical word onset, whereas violations of world knowledge, which result in implausible but still possible sentence meaning, did not differ from plausible sentences in this time window. Similar results were shown by *Kuperberg et al. (2003)*, *Geyer et al. (2006)* and *Paczynski et al. (2006)*. When *Kuperberg (2007)* evaluated factors evoking a late positivity, she concluded that none of the following factors—the presence of selection-restriction violations, semantic associations between the critical word and the preceding context, specific task instructions, or constraining context—by themselves could explain all results. One hypothesis that she advanced was that the impossibility to establish an overall meaning for the sentence might be the crucial factor inducing a late positivity on the critical word.

Given the variable nature of late positivities, and given the dire need for replication studies of phenomena that are rather new or for which evidence is scarce (see *Nieuwland et al., 2017*; see also *Dennis & Valacich, 2014*), the present study aimed to replicate *DeLong, Quante & Kutas (2014)* second experiment, with German stimuli presented to German native speakers. In addition, inspired by suggestions made by *Kuperberg (2007)* and *DeLong, Quante & Kutas (2014)*, we analyzed differences between implausible word completions that resulted in either possible or impossible overall

sentence meaning. To create a condition of impossible sentence meaning, we divided the materials into impossible and possible sets by means of subjective possibility ratings, collected in a pretest.

Following *DeLong, Quante & Kutas (2014)*, we predicted a graded effect of contextual fit of critical words at the level of the N400, with implausible continuations showing enhanced negativity relative to unexpected but plausible words. Next, we expect a predictability effect, showing as an anterior late positivity—relative to expected nouns—to unexpected but plausible sentence completions, but not to implausible nouns. Next, we predict a plausibility effect, with a posterior positivity only for implausible, anomalous sentence completions. If *Kuperberg's (2007)* assumption is correct, we predict this posterior late positivity only for those sentence completions that are truly anomalous and lead to an impossible overall sentence meaning, but not for those that allow an integration of the critical word with the preceding discourse, resulting in a perhaps implausible but nevertheless possible real-world meaning. This would constitute an effect of possibility, which also might show in a difference between possible and impossible implausible continuations in late positivity at anterior sites, with the possible continuations coinciding with plausible but unexpected ones.

## MATERIAL AND METHODS

### Stimuli

Stimuli were 150 constraining German sentence pairs (mean contextual constraint = 0.77, SD = 0.14, see cloze probability norming described below), which led to expectations for particular sentence-medial words. Following the condition labels used in *DeLong, Quante & Kutas (2014)*, each of the 150 contexts was completed by (a) the semantically expected noun (with the highest cloze probability for the specific context; EXP), (b) an unexpected but somewhat plausible noun (USP) and (c) an unexpected, implausible noun (ANOM), resulting in a total of 450 sentence pairs (see Table 1 for sample sentence pairs; the complete set of sentence pairs is provided in Table S1). To investigate whether the possible construction of overall sentence meaning was crucial for late positivities, the materials in the ANOM condition were subdivided on the basis of a pretest. Some of the sentences pairs in the ANOM condition contained critical nouns that allowed for a possible real-life meaning (ANOM-Pos; 45 sentence pairs), the other sentence pairs did not (ANOM-Impos; 105 sentence pairs). A total of 50 additional moderately constraining sentence pairs completed by their expected critical noun were used as fillers to balance the proportion of sentence pairs completed by expected vs unexpected nouns. Sentence material were either German translations of stimuli used in *DeLong, Quante & Kutas (2014)* or constructed in the same fashion by the experimenters. Where possible, critical nouns of the expected condition were re-used with different sentences in the other two conditions (53.3% of critical words were used three times, 31.1% were used twice and 15.6% were used only once). Since German nouns are coded for gender (masculine, feminine, neutral), all three completions of a particular sentence pair had the same grammatical gender. Written word frequency, word length and orthographic neighborhood size of the critical nouns were matched between the three main conditions

**Table 1  Sample sentence pairs.**

**EXP, USP, ANOM-Impos**

1. Peter stand bei Morgendämmerung auf, fuhr den ganzen Tag Traktor und fütterte abends seine Kühe. An manchen Tagen wäre er aber lieber kein [Bauer, Erwachsener, Trick] sondern ein unbekümmertes Kind. (*Peter gets up at dawn, drives the tractor all day and feeds his cows in the evening. On some days he would rather not be a [farmer, adult, trick] but a carefree child.*)

*Comprehension question:* Does Peter have cows?

2. Alice brach sich ihr Bein im Wanderurlaub. Der Arzt röntge ihr Bein und legte es in einen [Gips, Rollstuhl, Vogel] für zehn Wochen. (*Alice broke her leg while hiking. The doctor x-rayed her leg and put it in a [cast, wheelchair, bird] for 10 weeks.*)

3. Anne schrieb gerade ihre Masterarbeit und brauchte noch weitere Quellen für ihre Annahmen. Deshalb machte sie sich auf den Weg in eine [Bibliothek, Lehrbuchsammlung, Feder] für ihren Fachbereich. (*Anne was writing her master's thesis and needed more sources for her assumptions. Therefore, she made her way to a [library, textbook collection, feather] for her department.*)

**EXP, USP, ANOM-Pos**

4. Luisas neues WG-Zimmer war sehr klein, hatte aber hohe Decken. Um Platz zu sparen, kaufte sie sich deshalb ein [Hochbett, Aufbewahrungssystem, Schwein] im Baumarkt. (*Luisa's new room was very small but had high ceiling. To save space, she bought herself a [loft bed, storage system, pig] in the store.*)

*Comprehension question:* Was Luisa's new room very small?

5. Frank hält sich selbst für einen Komiker. Trotzdem kennt er nicht einen [Witz, Schauspieler, Anzug] oder Sketch, über den sein Publikum lachen würde. (*Frank considers himself quite a comedian. But he doesn't know a [joke, actor, suit] or sketch his audience would laugh about.*)

6. Marleen war schüchtern und konnte nicht gut mit Lob umgehen. Sie war peinlich berührt durch ein [Kompliment, Tattoo, Bügeleisen] ihres Vorgesetzten. (*Marleen was very shy and could not handle praise well. She was embarrassed by a [compliment, tattoo, iron] from her supervisor.*)

**FILL**

7. Marina war viel auf Reisen und erlebte fast jeden Tag etwas Neues. Um sich an alles zu erinnern, schrieb sie ein [Tagebuch] und klebte Fotos dazu. (*Marina travels a lot and has new experiences almost every day. To remember everything, she writes a [diary] and adds pictures.*)

(see Table 2). Note that the Pos and Impos items within the ANOM condition were not balanced with respect to these factors.

## Cloze probability norming

Stimulus norming for critical noun cloze probability was conducted in a separate sentence completion task with 36 volunteers (native speakers of German, mainly students). They were compensated with course credit and did not participate in the EEG study. Contexts were truncated prior to the critical noun, and participants were asked to complete the second sentence with a single noun that came to their mind first and fitted with the preceding context. Every context ended with the three German indefinite articles, in the order masculine, feminine and neuter, thus allowing nouns of any grammatical gender. Cloze probability was calculated as the proportion of participants who completed a particular sentence pair with a particular noun. The cloze probability of the most frequent noun equals the contextual constraint of a given sentence pair. Sentence pairs with a cloze probability of 50% or higher were considered highly constraining and included in the study. Filler sentences had a cloze probability of 40% or higher. Table 2 shows mean cloze probabilities for the experimental and filler conditions.

## Plausibility rating

All 450 sentence pairs, truncated after the critical noun, were rated for plausibility ("How plausible is the sentence pair's meaning . . . ") on a scale of 1 (not plausible) to 5 (highly plausible) by eight independent German raters who did not participate in

**Table 2  Stimuli characteristics.**

| Condition label | Condition | Number of items | Mean critical noun cloze probability (SD), Range: 0–1 | Mean context + noun plausibility rating (SD), Range: 1–5 | Mean context + noun possibility rating (SD), Range: 1–4 | Mean contextual constraint (SD), Range: 0–1 | Mean critical noun written frequency (SD)[a] | Mean critical noun length (SD) | Mean critical orthographic neighborhood size (SD)[b] |
|---|---|---|---|---|---|---|---|---|---|
| EXPected | High cloze/High plausibility | 150 | 0.77 (0.14) | 4.68 (0.36) | 3.77 (0.33) | 0.77 (0.14) | 2,148.27 (4,233.85) | 6.91 (2.55) | 11.99 (16.08) |
| Unexpected somewhat plausible (USP) | Low cloze/High plausibility | 150 | <0.01 (<0.01) | 2.96 (0.99) | 3.19 (0.56) | 0.77 (0.14) | 2,551.02 (5,749.39) | 7.38 (3.15) | 12.76 (17.99) |
| ANOMalous | Low cloze/Low plausibility | 150 | <0.01 (<0.01) | 1.05 (0.13) | 1.44 (0.45) | 0.77 (0.14) | 2,278.32 (4,278.45) | 6.95 (2.75) | 12.89 (18.00) |
| ANOM-Impos | ANOM + impossible meaning | 105 | <0.01 (<0.01) | 1.02 (0.11) | 1.20 (0.18) | 0.77 (0.14) | 2,659.35 (4,840.02) | 6.57 (2.45) | 14.89 (19.81) |
| ANOM-Pos | ANOM + possible meaning | 45 | <0.01 (<0.01) | 1.11 (0.14) | 2.00 (0.38) | 0.78 (0.14) | 1,369.05 (2,268.36) | 7.84 (3.02) | 8.14 (11.53) |
| FILL | High cloze/High plausibility | 50 | 0.76 (0.18) | – | – | 0.76 (0.18) | 3,276.12 (5,290.63) | 7.20 (2.86) | 9.62 (11.36) |

**Notes:**
[a] Absolute annotated type frequencies according to dlexDB (http://www.dlexdb.de/).
[b] Orthographic neighborhood size (as defined by *Coltheart et al., 1977*) according to dlexDB.

**Table 3 Differences between conditions.**

| Comparison | Plausibility | Possibility | Word frequency | Orthographic neighbors | Word length | Contextual constraint |
|---|---|---|---|---|---|---|
| EXP vs USP | $t(298) = 19.89$, **$p < 0.001$** | $t(298) = 10.81$, **$p < 0.001$** | $t(298) = -0.63$, $p = 0.528$ | $t(298) = -0.30$, $p = 0.765$ | $t(298) = -1.43$, $p = 0.154$ | – |
| EXP vs ANOM | $t(298) = 116.21$, **$p < 0.001$** | $t(298) = 51.10$, **$p < 0.001$** | $t(298) = -0.26$, $p = 0.792$ | $t(298) = -0.45$, $p = 0.650$ | $t(298) = -0.15$, $p = 0.879$ | – |
| USP vs ANOM | $t(298) = 23.47$, **$p < 0.001$** | $t(298) = 29.96$, **$p < 0.001$** | $t(298) = 0.41$, $p = 0.684$ | $t(298) = -0.15$, $p = 0.882$ | $t(298) = 1.25$, $p = 0.213$ | – |
| ANOM-Pos vs ANOM-Impos | $t(67.58) = 3.88$, **$p < 0.001$** | $t(53.11) = 13.62$, **$p < 0.001$** | $t(145.15) = -2.21$, **$p = 0.028$** | $t(131.79) = 2.60$, **$p = 0.011$** | $t(67.00) = 2.38$, **$p = 0.020$** | $t(85.06) = 0.34$, $p = 0.738$ |

**Notes:**
Because of unequal group sizes, a Welch-test was conducted in case of ANOM-Pos vs ANOM-Impos.
Significant $p$-values are marked in bold.

the EEG study. Table 2 presents mean plausibility ratings of all conditions. Following *DeLong, Quante & Kutas (2014)*, mean plausibility was >1.5 in the EXP and USP conditions, and ≤1.5 in the ANOM condition. The plausibility ratings differed significantly between all conditions (see Table 3).

## Possibility rating

All 450 sentence pairs, truncated after the critical noun, were rated for possibility ("How possible (in real-life) is the sentence pair's meaning...") by the same eight raters, on a scale of 1 (impossible) to 4 (possible). Table 2 specifies mean possibility ratings for all conditions. Similar to the cut-off for Plausibility, mean possibility ratings were >1.5 in the EXP, USP and ANOM-Pos conditions, but ≤1.5 in the ANOM-Impos condition. The ratings of all conditions differed significantly from each other (see Table 3). Participants performed both plausibility and possibility ratings at the same time. No examples were provided to avoid biasing the raters' judgments. Correlations between plausibility and possibility ratings are displayed in Analysis S1.

In the main experiment, each participant was presented with one of three 200-item lists, with contexts and critical nouns used once per list (except for four critical nouns that occurred twice per list, in different contexts). Lists 1, 2 and 3 were presented to 12, 10 and 10 participants, respectively. Every list consisted of 50 predictable, expected nouns, 50 unexpected plausible nouns, 50 unexpected implausible (ANOM) nouns and 50 fillers. Approximately one third of the ANOM nouns was rated possible (list 1: 17, list 2: 16, list 3: 12), the remaining two thirds were rated impossible. A total of 50 comprehension questions followed 25% of sentence pairs at random intervals. Three additional sentence pairs preceded every list to familiarize participants with the task. Sentence pairs within a list were randomized across subjects.

## ERP participants

A total of 32 students (23 f, nine m) participated in the experiment after giving written informed consent. They were compensated with course credit or cash (7.50 €/h). Mean age was 25.3 years (19–34). All participants were monolingual native speakers

of German and right-handed (assessed via Edinburgh Handedness Inventory, *Oldfield, 1971*). Eight participants reported a left-handed parent or sibling, one reported two left-handed relatives. All participants reported normal or corrected-to-normal vision. One additional participant was tested but excluded from analysis because of a technical problem during the experiment. The study protocol was conducted in accordance with ethical standards of the Declaration of Helsinki and approved by the local ethics committee of the University of Münster (approval number #2016-42-LQ).

## Procedure

The experiment consisted of a single 2-h-session conducted in a quiet and dimly lit room at the Westfälische Wilhelms-Universität Münster. Participants were seated approximately one m in front of a LED monitor (BenQ, model XL2420T, 144 Hz, 24″W) and read sentence pairs for comprehension. The experiment was set up using Presentation software (NeuroBehavioral Systems, Version 16.3). Stimuli were presented visually, in black type (RGB: 0, 0, 0; Arial 48 pt) on a gray background (RGB: 148, 148, 148). The experiment was divided into eight blocks of approximately 6 min length, with 2 min breaks between blocks. Every trial started with a fixation cross (500 ms) in the center of the screen, followed by the first sentence of a pair presented in its entirety. Participants advanced to the critical sentence via button press. This sentence including the critical word was presented with a rapid serial visual presentation technique, each word presented centrally for 200 ms, with a stimulus onset asynchrony of 500 ms. Yes/no comprehension questions followed 25% of sentence pairs at random intervals. Participants responded with two buttons on a response pad (Cedrus, model RB-830) with response buttons counterbalanced across participants and lists. Comprehension questions appeared after the critical noun sentence. In case of a question, participants' button press advanced to the next trial, otherwise the next sentence pair appeared automatically after 2 s.

Material, design and procedure were almost identical to DeLong et al.'s second experiment except for the following differences. First, DeLong et al.'s sentence material was translated into German or constructed using the same sentence structure. Second, mean constraint of discourse contexts and mean cloze probability of expected critical words were lower than in DeLong et al. (0.77 vs 0.89, respectively). Third, sentence pairs were not rated for possibility in DeLong's study. Fourth, 32 participants completed the present EEG experiment, 24 students participated in DeLong et al.'s second study.

## Electroencephalographic recording parameters

Electroencephalography was recorded from 32 Ag/AgCl-electrodes attached to a WaveGuard 32-channel cap Advanced Neuro Technology (ANT). Electrodes were placed according to the International 10–20 convention (*Jasper, 1958*), and an average reference was used (see Fig. 1 for scalp sites). Blinks and vertical eye movements were monitored from electrodes placed above and below the left eye, and horizontal eye movements were monitored from two electrodes placed on the outer canthi. Impedances were kept below five kΩ. The EEG was continuously recorded with Advanced Source Analysis (version 4.7.3.1, ANT). Data collection and evaluation were controlled by ExMan

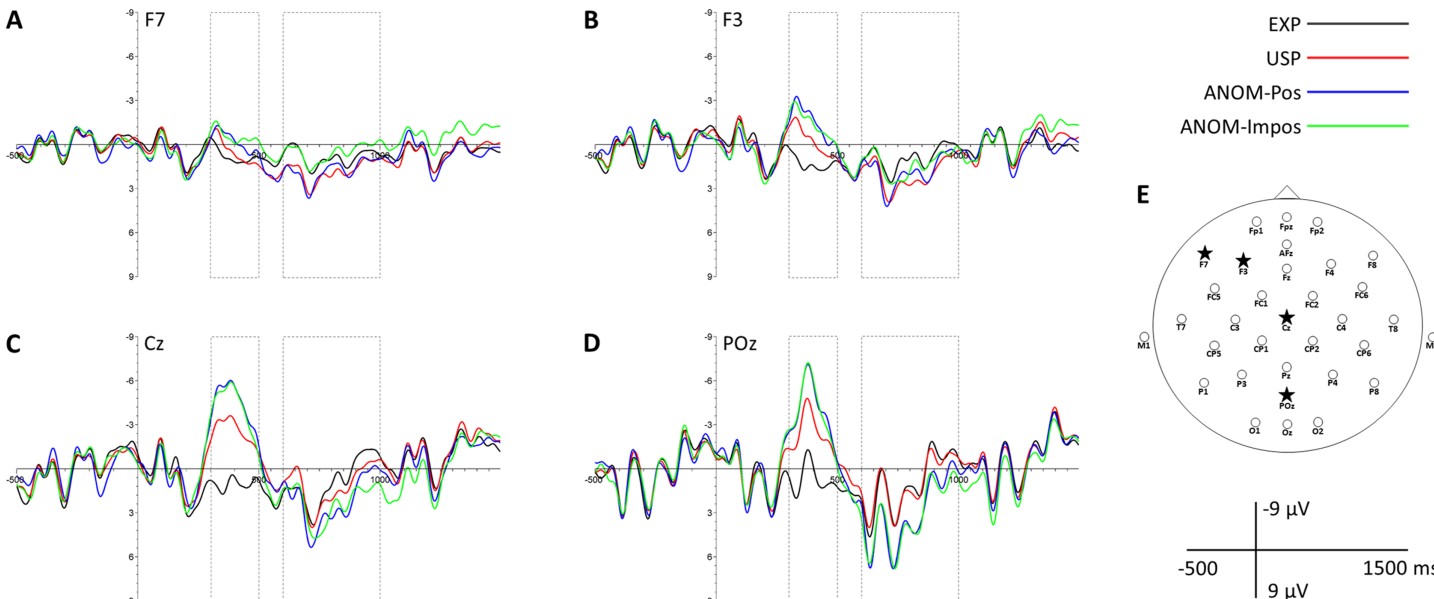

**Figure 1 Representative anterior and posterior scalp channels.** ERPs of channels F7 (A), F3 (B), Cz (C) and POz (D) for EXP, USP, ANOM-Pos and ANOM-Impos nouns. Displayed channels are marked as stars on the electrode montage mapping (E). Dashed-line boxes indicate analyzed time windows (N400: 300–500 ms; post-N400 positivity: 600–1,000 ms).

(Experiment Manager; MS Excel worksheet with active macros). EEG was amplified (ExG 20×, fixed = 50 mV/V), low pass filtered (finite impulse response filter, cut-off frequency = 0.27 × sampling rate) and continuously digitized at a sampling rate of 256 samples/s.

## Data analysis

Before averaging, the EEG signal was filtered using a Butterworth half-amplitude bandpass FIR-filter (0.1 Hz, 20 Hz, 12 db/oct). Vertical eye movements were corrected with principal component analysis (*Ille, Berg & Scherg, 2002*). Additionally, seven electrodes (0.7%) were interpolated (see Table S2). The EEG was re-referenced offline to the algebraic mean of left and right mastoids and averaged for each experimental condition, time-locked to the critical noun onset. Before averaging, trials contaminated by artefacts (specified as voltage changes exceeding ±75 μV during the epoch) were rejected offline (on average 4.23% of all trials, SD = 4.80). ERPs were calculated for epochs extending from 500 ms pre- to 1,500 ms post-stimulus onset, thus using a pre-stimulus baseline of 500 ms.

In a first step, mass univariate analyses were conducted to compare spatial and temporal properties of possible ERP effects found in the present experiment to the findings by *DeLong, Quante & Kutas (2014)*. ERPs from the three pairwise comparisons, [USP minus EXP], [ANOM minus EXP], and [ANOM minus USP], were submitted to repeated measures, two-tailed *t*-tests at all sampled time points between 250 and 1,050 ms (206 total time points) at all 30 scalp electrodes, resulting in 6,180 total comparisons for each condition contrast. To control the number of false discoveries, the *Benjamini & Yekutieli (2001)* procedure was applied using a false discovery rate level of 5%.

In a second step, mean amplitudes were analyzed by first conducting ANOVAs with three levels of noun type (EXP, USP and ANOM) to compare the present results directly to those reported in *DeLong, Quante & Kutas (2014)*. These analyses were complemented by pairwise *t*-tests between the four levels of noun type (EXP, USP, ANOM-Pos and ANOM-Impos). ANOVAs were applied to the data from three time windows: (a) over all 30 electrode sites between 300 and 500 ms (N400), (b) over seven (left) anterior electrode sites [Fp1, Fpz, F7, F3, Fz, FC5, T7] between 600 and 1,000 ms (frontal positivity), (c) over seven posterior electrode sites [Cz, CP1, CP2, P3, Pz, P4, POz] between 600 and 1,000 ms (posterior positivity; see Fig. 1 for electrode placement). Scalp regions and temporal windows were based on *DeLong, Quante & Kutas (2014)* second experiment. To confirm the left lateralization of the anterior positivity, we extended the corresponding ANOVA by the factor hemisphere (left, right), and included equivalent right hemisphere electrodes (Fp2, F4, F8, FC6 and T8) while excluding midline electrodes Fz and Fpz. If sphericity was violated, ANOVA *p*-values and degrees of freedom were corrected using epsilon correction (Greenhouse Geisser) for repeated measures with more than one degree of freedom. Significance levels of pairwise *t*-tests were Bonferroni-adjusted (*t*-tests following ANOVAs with three levels of noun type: $p_{boncor} < 0.0167$, *t*-tests comparing all four levels of noun type: $p_{boncor} < 0.0083$).

# RESULTS

## Behavioral results

Participants correctly answered an average of 96.7% (median 97%, range = 90–100%) of yes/no comprehension questions, suggesting they comprehended the sentence pairs during the experiment.

## ERP results

### Mass univariate analyses

The first mass univariate analysis focused on predictability, comparing USP vs EXP nouns (see Fig. 2A). There was a widespread N400 effect, with ERPs to USP nouns being more negative than ERPs to EXP nouns. This negativity lasted from approximately 250 to 500 ms. Starting shortly before the offset of the N400, between approximately 550 and 1,000 ms, USP nouns were more positive than EXP nouns, particular over left frontal and left lateral temporo-parietal scalp sites.

The second mass univariate analysis looked at plausibility, comparing ANOM vs EXP nouns (see Fig. 2B). Again, a widespread N400 effect emerged, with ERPs to ANOM nouns being more negative than ERPs to EXP nouns, between approximately 250 and 500 ms. By about 600 ms, a positivity of ANOM relative to EXP nouns emerged and continued up to the end of the time window (1,050 ms), being most prominent over central and posterior scalp sites.

The third mass univariate analysis compared ANOM vs USP nouns (see Fig. 2C). Between 300 and 450 ms, ERPs to ANOM nouns were more negative than ERPs to USP nouns. In addition, from approximately 600 ms to the end of the time window (1,050 ms), ERPs to ANOM nouns were more positive than ERPs to USP nouns at

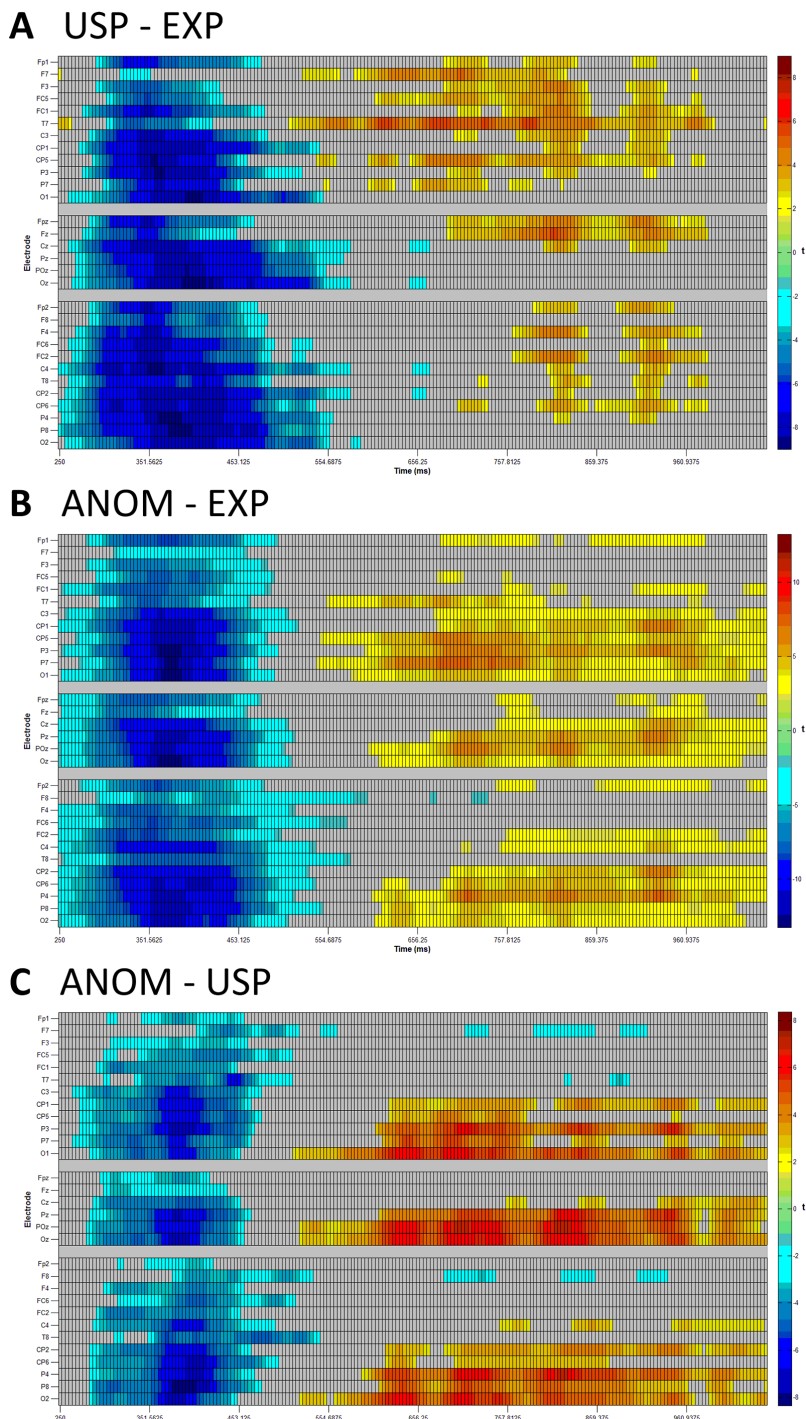

**Figure 2 Mass univariate analyses.** Raster plots of *t*-values with control for false discovery rates in two-dimensional grids of the following comparisons: (A) USP nouns minus EXP nouns, (B) ANOM nouns minus EXP nouns and (C) ANOM nouns minus USP nouns. Results are plotted in four millisecond lags. Left scalp electrodes are displayed in the upper section, midline scalp electrodes in the center and right scalp electrodes in the lower section of each panel. Red (blue) indicates that ERPs to the first noun type are more positive (negative) than ERPs to the second noun type. See Fig. 1E for electrode placement.

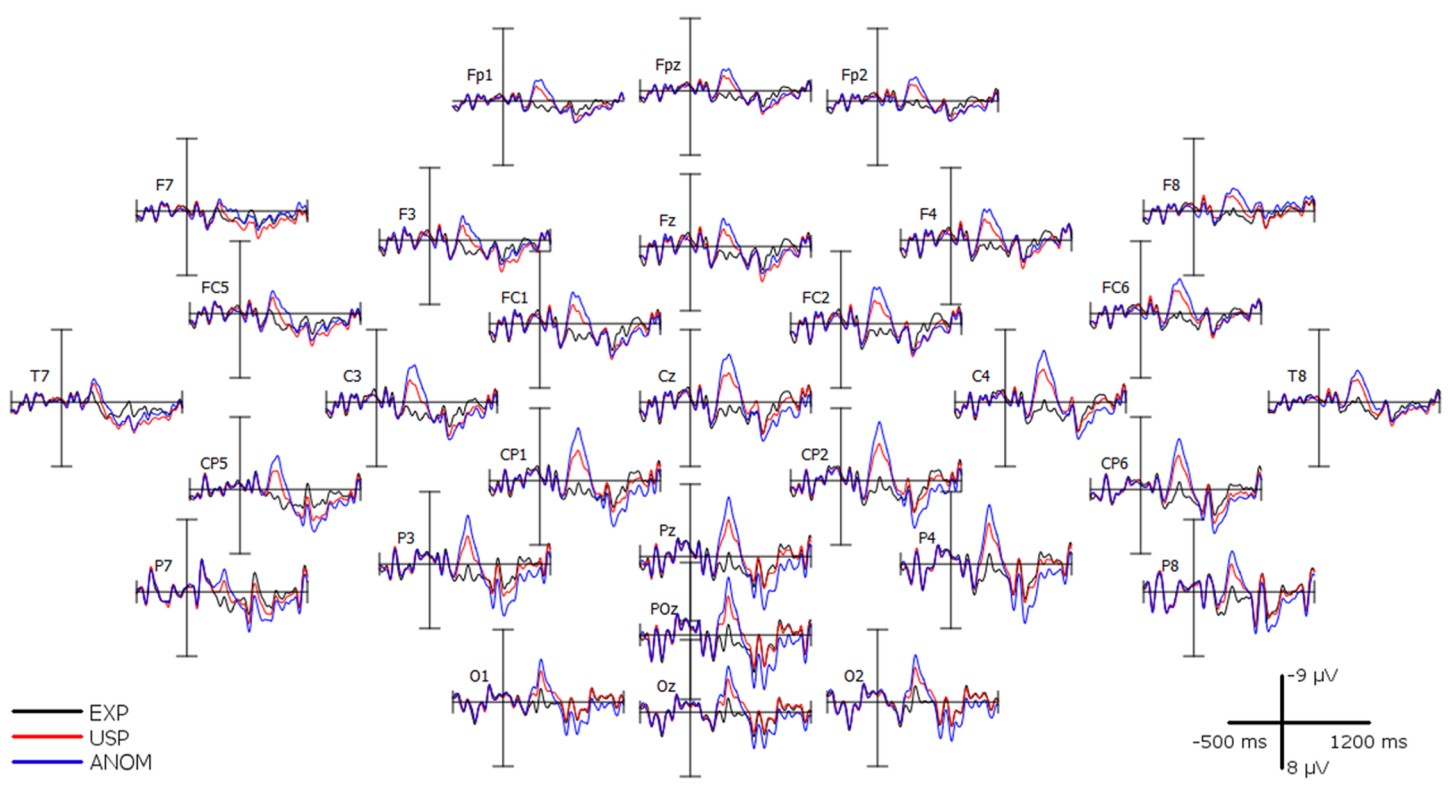

**Figure 3 Grand average (*n* = 32) recorded over 30 scalp channels.**

posterior scalp locations. In contrast, ERPs to USP nouns were more positive than ERPs to ANOM nouns over lateral frontal scalp sites between approximately 700 and 900 ms. In all three mass-univariate analyses, significant *p*-values are $p_{adj} < 0.05$.

### Analyses of variance

For visual inspection, Fig. 3 shows the grand average ERPs of all 32 participants over 30 scalp channels. Topographic scalp maps of ERP mean amplitude voltage differences can be seen in Fig. 4, and four representative anterior and posterior channels are shown in Fig. 1. In line with the results from mass univariate analyses described above, all figures reveal N400 effects for both USP and ANOM nouns relative to EXP nouns, a post-N400 positivity for ANOM nouns over posterior channels, and a post-N400 positivity for USP nouns over anterior channels. Early components (P1, N1 and P2) do not differ as a function of noun type. Tables 4 and 5 provide mean amplitudes of the four noun types and detailed results of pairwise *t*-tests between conditions.

### 300–500 ms

An ANOVA with three levels of noun type over all 30 electrode sites revealed a main effect [$F(1.60, 49.46) = 74.98$, $p < 0.001$, $\varepsilon_{GG} = 0.80$, $\eta_p^2 = 0.39$]. ANOM nouns showed the largest negativity ($-2.89\ \mu V$), followed by USP nouns ($-1.54\ \mu V$) and EXP nouns ($0.83\ \mu V$). Post hoc *t*-tests are displayed in Fig. 5. The pairwise *t*-tests between all four noun types revealed significant differences between all conditions ($t(31) \geq 4.38$,

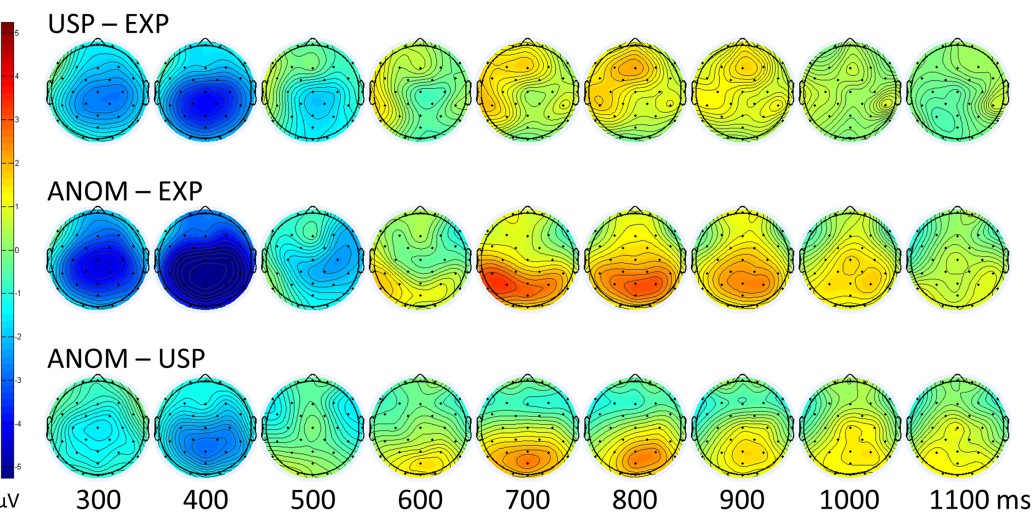

**Figure 4 Topographic scalp maps.** ERP mean voltage differences of the three main comparisons for time points 300–1,100 ms.

**Table 4 Mean amplitude and standard deviation (μV) of the four noun types across time windows and different scalp sites.**

|                     | EXP         | USP          | ANOM         | ANOM-Pos     | ANOM-Impos   |
| ------------------- | ----------- | ------------ | ------------ | ------------ | ------------ |
| N400                | 0.83 (1.63) | −1.54 (1.82) | −2.89 (2.32) | −3.01 (2.41) | −2.82 (2.43) |
| Anterior positivity | 0.79 (1.24) | 1.91 (1.32)  | 1.51 (2.05)  | 1.96 (2.19)  | 1.33 (2.27)  |
| Posterior positivity| 0.98 (1.68) | 1.38 (1.66)  | 3.09 (2.32)  | 2.95 (2.42)  | 3.16 (2.46)  |

$p < 0.001$) except for the comparison of ANOM-Pos (−3.01 μV) and ANOM-Impos nouns (−2.82 μV; $t(31) = −0.72$, $p = 0.479$).

*600–1,000 ms posterior scalp sites*
An ANOVA with three levels of noun type, conducted over seven posterior electrode sites, showed a main effect [$F(2, 62) = 21.06$, $p < 0.001$, $\eta_p^2 = 0.19$]. ANOM nouns had the largest positivity (3.09 μV), followed by USP nouns (1.38 μV) and EXP nouns (0.98 μV). Post hoc $t$-tests are displayed in Fig. 5. Pairwise $t$-tests revealed significant differences between EXP and ANOM-Pos nouns ($t(31) = −4.29$, $p < 0.001$), EXP and ANOM-Impos nouns ($t(31) = −5.40$, $p < 0.001$), USP and ANOM-Pos nouns ($t(31) = −4.75$, $p < 0.001$) and USP and ANOM-Impos nouns ($t(31) = −4.87$, $p < 0.001$). No reliable difference was found between EXP and USP nouns ($t(31) = −1.31$, $p = 0.200$) and between the two types of ANOM nouns (ANOM-Pos = 2.95 μV, ANOM-Impos = 3.16 μV; $t(31) = −0.72$, $p = 0.477$).

*600–1,000 ms anterior scalp sites*
The extended ANOVA indicated a left lateralization of the effect (see Analysis S2). Therefore, we restricted our analysis to those electrodes analyzed in *DeLong, Quante & Kutas (2014)*. Over the seven left anterior electrodes, the ANOVA with three levels of noun type revealed a main effect [$F(2, 62) = 6.13$, $p = 0.004$, $\eta_p^2 = 0.08$]. USP nouns showed

**Table 5  Pairwise *t*-tests between the four noun types.**

|  | Mean of the differences [μV] | *t*(31) | *p* | 95% confidence interval |
|---|---|---|---|---|
| **300–500 ms, all scalp sites (N400)** |  |  |  |  |
| EXP vs USP | 2.37 | 8.48 | <0.001* | [1.80; 2.94] |
| EXP vs ANOMI | 3.65 | 9.26 | <0.001* | [2.85; 4.46] |
| EXP vs ANOMP | 3.84 | 9.83 | <0.001* | [3.04; 4.64] |
| USP vs ANOMI | 1.28 | 4.38 | <0.001* | [0.68; 1.88] |
| USP vs ANOMP | 1.47 | 5.83 | <0.001* | [0.95; 1.98] |
| ANOMP vs ANOMI | −0.19 | −0.72 | 0.479 | [−0.72; 0.35] |
| **600–1,000 ms, posterior scalp sites** |  |  |  |  |
| EXP vs USP | −0.40 | −1.31 | 0.200 | [−1.02; 0.22] |
| EXP vs ANOMI | −2.17 | −5.40 | <0.001* | [−2.99; −1.35] |
| EXP vs ANOMP | −1.97 | −4.29 | <0.001* | [−2.90; −1.03] |
| USP vs ANOMI | −1.77 | −4.87 | <0.001* | [−2.52; −1.03] |
| USP vs ANOMP | −1.57 | −4.75 | <0.001* | [−2.24; −0.89] |
| ANOMP vs ANOMI | −0.21 | −0.72 | 0.477 | [−0.80; 0.38] |
| **600–1,000 ms, anterior scalp sites** |  |  |  |  |
| EXP vs USP | −1.12 | −3.97 | <0.001* | [−1.70; −0.55] |
| EXP vs ANOMI | −0.54 | −1.52 | 0.138 | [−1.27; 0.18] |
| EXP vs ANOMP | −1.17 | −2.88 | 0.007* | [−2.00; −0.34] |
| USP vs ANOMI | 0.58 | 1.50 | 0.145 | [−0.21; 1.37] |
| USP vs ANOMP | −0.05 | −0.12 | 0.904 | [−0.84; 0.74] |
| ANOMP vs ANOMI | 0.63 | 1.89 | 0.068 | [−0.05; 1.31] |

**Note:**
  * Significant after Bonferroni adjustment ($p_{\text{boncor}} < 0.0083$).

the greatest positivity (1.91 μV), followed by ANOM nouns (1.51 μV) and EXP nouns (0.79 μV). Post hoc *t*-tests are displayed in Fig. 5. Paired *t*-tests revealed that EXP nouns differed from USP nouns ($t(31) = -3.97$, $p < 0.001$) and from ANOM-Pos nouns ($t(31) = -2.88$, $p = 0.007$). The differences between USP and ANOM-Pos nouns (diff = 0.05 μV; $t(31) = -0.12$, $p = 0.904$) and between EXP and ANOM-Impos nouns (diff = 0.54 μV; $t(31) = -1.52$, $p = 0.138$) were not significant. Although ANOM-Pos (1.96 μV) and ANOM-Impos nouns (1.33 μV) differed by 0.64 μV, this difference also did not reach significance ($t(31) = 1.89$, $p = 0.068$).

Whereas there were no differences in other temporal and/or spatial analysis windows, ANOM-Pos and ANOM-Impos thus seem to have a different impact on late anterior positivity. Given that different items were compared in the ANOM-Pos and ANOM-Impos conditions, we ran a regression analysis to assess possible effects of critical word characteristics that may cause amplitude differences between nouns. An amplitude calculation for each item averaged over participants is inadequate for exploring factors in multiple regression designs, because it disregards interparticipants' variability. Therefore, we used the method suggested by *Lorch & Myers (1990)*. For every participant, we extracted amplitudes of individual words and fitted a linear

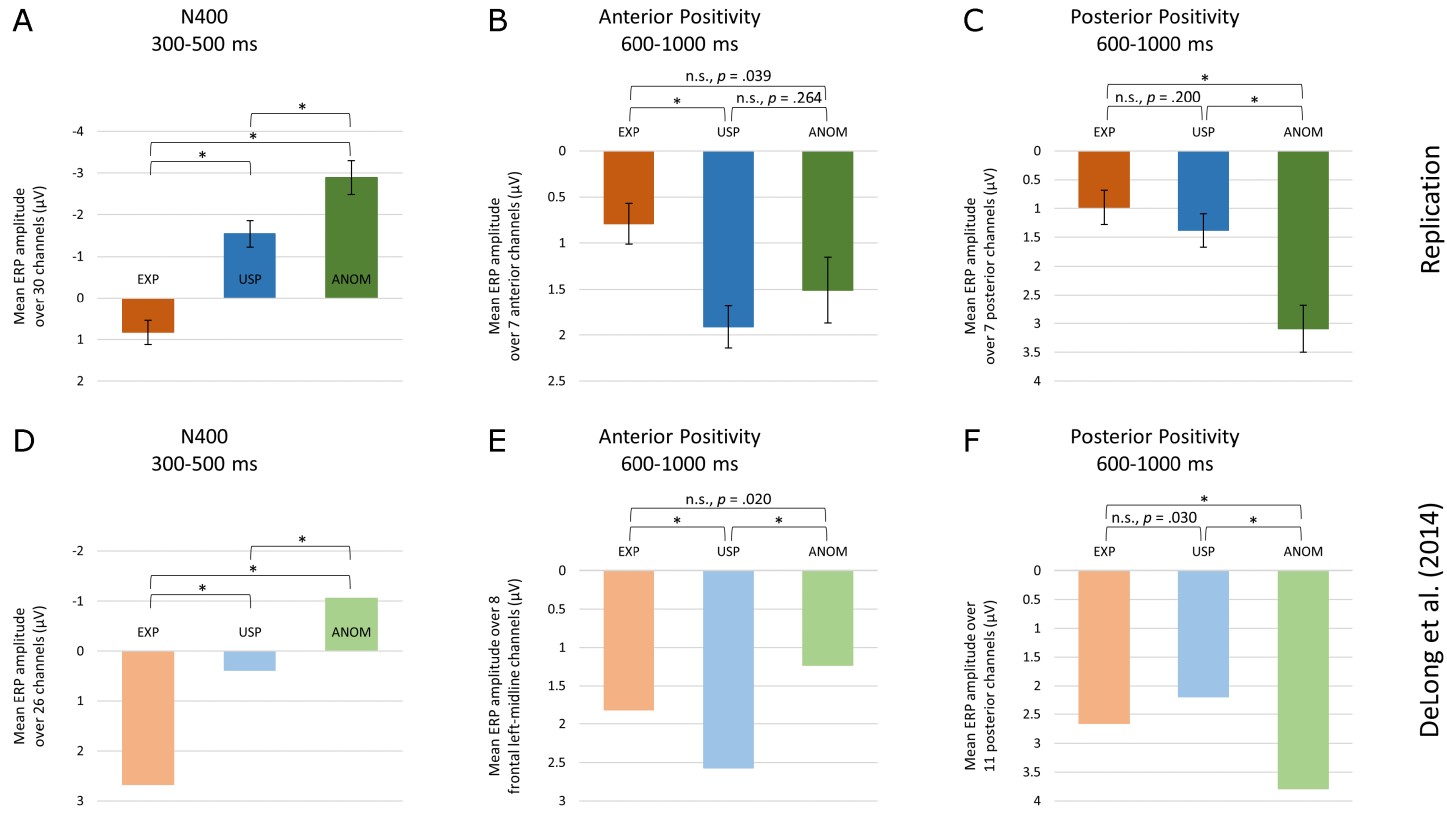

**Figure 5 Comparison of results.** EEG results from the current study (A–C) and Experiment 2 by *DeLong, Quante & Kutas (2014)* (D–F), for three time windows (N400, Anterior Positivity, Posterior Positivity) and three noun conditions (EXP, USP, ANOM). Significance levels of pairwise t-tests were Bonferroni-adjusted ($p_{boncor}$ < 0.0167).

regression with factors plausibility, possibility, word length, word frequency and orthographic neighborhood size (without interaction terms). For every predictor, the resulting 32 $t$-values entered a one-sample $t$-test. Only the effect of word length on amplitude was significantly different from zero ($t(31) = 3.25$, $p = 0.003$). See Table S3 for detailed results. Note that the effect of possibility failed significance, and even the plausibility effect, which entails a within-item comparison, is much weaker in this analysis.

## GENERAL DISCUSSION

In this study with German materials and participants, we investigated the electrophysiological signatures of different types of contextual fit—from predictable and thus expected, to highly implausible continuations of short discourses consisting of sentences pairs. Whereas no reliable differences were present before 300 ms, graded N400 effects for target nouns were observed as a function of their predictability and plausibility in the discourse. Relative to highly predictable nouns, amplitudes were more negative for unexpected but plausible continuations, and again more negative for implausible continuations. Whether or not the implausible noun was a somewhat strange but in principle possible continuation of the preceding discourse had no impact on the N400. In the time window following the N400, positivities with different scalp
signatures were observed that differed as a function of noun type. At posterior electrode sites, the 600–1,000 ms time window revealed similar amplitudes for highly predictable and unpredictable but plausible continuations. Relative to these two continuations, there was enhanced positivity for both implausible noun types—which showed very similar amplitudes. At anterior sites, predictable and completely impossible continuations had similar amplitudes, but predictable nouns had a less positive amplitude than both unpredictable plausible and implausible, but still possible continuations, which did not differ.

In the following, we compare our outcomes to the original study (Experiment 2) by *DeLong, Quante & Kutas (2014)* that we aimed to replicate, and evaluate our results against the predictions made for continuations that are quite implausible, but for which a real-world interpretation can be constructed given the discourse. We discuss the distinction between anterior and posterior late positivities and the potential processing functions that may underlie them.

## Replication of *DeLong, Quante & Kutas (2014)*

The data patterns relevant for the replication of *DeLong, Quante & Kutas (2014)*, with three conditions (EXP, USP and ANOM) in three time windows (N400, Anterior Positivity, Posterior Positivity), show a striking similarity between the two studies, illustrated in Fig. 5.

The data for the N400 from the two studies show a very similar pattern—ignoring the position of the zero line. Relative to expected continuations, negativity is enhanced for unexpected but plausible (USP) nouns and again more so for implausible (ANOM) nouns. The results for the N400 thus fully replicate the graded negativity reported by DeLong et al. Note that the possibility to create an interpretation for some implausible continuations had no effect on N400 amplitude, since our two anomalous conditions did not differ.

The posterior positivity after the N400 also shows the same pattern as obtained by *DeLong, Quante & Kutas (2014)*, with significant differences between the expected (EXP) and anomalous continuations (ANOM), between unexpected (USP) and anomalous words, but not between EXP and USP, the expected and unexpected continuations. Again, exactly the same pattern with the same significances was observed in both studies. Moreover, the analysis with four noun types showed no difference between the possible and impossible anomalous continuations. Finally, the anterior positivity again showed a similar pattern in both studies, but with somewhat different significances. Whereas in both studies, amplitudes for expected (EXP) and unexpected plausible (USP) nouns differ, and amplitudes for expected and anomalous nouns do not differ, the difference between the unexpected plausible and anomalous nouns that was reliable in DeLong et al. failed significance in our data. The analysis with four noun levels gives an indication why this might be the case. In this analysis, the anomalous nouns that have a possible interpretation given the preceding discourse do show a significant difference to the expected nouns, thus coinciding with the unpredictable but plausible nouns. The difference to the impossible anomalous nouns remains insignificant. Note however

that the post hoc regression analysis, which takes into account between-item differences in the analyses of possibility effects, questions whether these differences can be attributed to possibility.

Thus, with one interesting exception we closely replicate Experiment 2 by *DeLong, Quante & Kutas (2014)*, with German materials—mainly but not exclusively translated from *DeLong, Quante & Kutas (2014)*, with somewhat lower predictability of the predictable, expected nouns and with German native speakers. We believe this replication of a dissociation between anterior and posterior positivity in largely overlapping, post-N400 time windows to be an important contribution to the growing evidence for a functional difference associated with these two late positivities. As DeLong et al., and unlike other studies, we show this relatively new dissociation with the same population within one experiment. In the following, we discuss our findings relative to data, hypotheses and models proposed by others.

## N400 and late positivities

The N400 effects show that relative to an anomalous noun, an unexpected noun that is nevertheless a perfectly plausible continuation shows a smaller negativity. This graded negativity, relative to the predicted continuation, replicates findings from many studies that show amplitude negativity to depend on the degree of deviation from the condition that serves as reference (see *Kutas & Federmeier, 2011*, for an overview).

The late positivity observed in our data seems to come in two guises. There is a bilateral, posterior positivity that separates expected and unexpected but plausible continuations from implausible, anomalous continuations—with no difference between those for which a possible, real-world meaning (ANOM-Pos) can be constructed and those for which this is not the case (ANOM-Impos). A second late positivity, with anterior, left-lateralized scalp distribution, seems to distinguish between nouns for which an interpretation in the given discourse is possible but unexpected (USP and ANOM-Pos nouns) on the one hand, and predictable words on the other.

## Posterior late positivity

The posterior late positivity observed in our data resembles the P600 that has been commonly associated with syntactic violations (*Hagoort, Brown & Groothusen, 1993*) or syntactic complexity (*Friederici, Hahne & Saddy, 2002*; *Kaan & Swaab, 2003*). This changed some 15 years ago, when late positivities were reported for words that constituted thematic-role violations (e.g., *At breakfast the eggs would eat*) which are semantic in nature (*Kolk et al., 2003*; *Kuperberg et al., 2003*; *Hoeks, Stowe & Doedens, 2004*; see *Brouwer, Fitz & Hoeks, 2012*, for an overview). Such "semantic illusions" had no impact on the N400 but showed in late positivities, with a central-posterior/parietal scalp topography that resembles the "syntactic" P600. This "semantic" P600 again fired the debate on its functional significance. Most proposed models and views adhere to two processing streams—semantic and syntactic—whose outputs can conflict with each other, which is reflected in the P600 (cf. *Kim & Osterhout, 2005*; *Kolk & Chwilla, 2007*; *Kuperberg, 2007*;

*Bornkessel-Schlesewsky & Schlesewsky, 2008*; *Hagoort, Baggio & Willems, 2009*; *Kos et al., 2010*; *Metzner et al., 2017*; see *Brouwer et al., 2017*, for an overview and a different model).

In their seminal review of data from about 60 studies, *Van Petten & Luka (2012)* conclude that posterior late positivity is associated with attempts at reanalysis when a problem is detected—be it a syntactic or semantic incongruency or anomaly. Our late posterior positivities for all anomalous continuations fit this picture. *Kuperberg (2013)* prominently put late positivities into the perspective of prediction, suggesting that the posterior late positivities reflect processing costs when the incoming word disconfirms predicted events or event structure. This is the case even for semantic illusions (e.g., *The cat that from the mice fled*, incoming word underlined, *Kolk et al., 2003*) in which the incoming words semantically fit the event, but their thematic roles violate event structure. In our data, all anomalous continuations show a posterior negativity. Clearly, impossible continuations violate event structure (often but not always because of selection restriction violations): "excuse" is not a viable candidate for a snowman's nose. This is different for the unexpected but plausible continuations: lacking a carrot, a banana can serve as a snowman's nose, and is thus compatible with the event of snowman construction. Consequently, our expected and unexpected but plausible words do not differ in late posterior positivity. Note that relative to these two conditions, and in contrast to our prediction, a clear late posterior positivity was evident for both types of anomalous continuations, those that are completely impossible and those for which an admittedly strange meaning could be constructed (e.g., *To save space, she bought herself a pig (expected: a loft bed) in the store.*). Following the logic by Van Petten and Luka, both anomalies initiate the reprocessing of prior input, and in Kuperberg's view, both anomalies seem severe enough to violate event structure.

## Anterior late positivity

Finally, we consider the anterior post-N400 positivity observed in our data. In the overall analysis, unexpected but plausible continuations (*the banana as nose for the snowman*) and implausible but still in some way possible continuations (*the woman who bought herself a pig to save space*) group together. First, they both differ from expected, highly predictable continuations (*the carrot for a snowman's nose, a loft bed to save space*) and second, both continuations allow for a revision of the discourse on the basis of the meaning of the unexpected words. Note that the differences observed here may be due to item characteristics, as the regression analysis indicated. Although these data, given that they involve different items, should be treated with caution, it is interesting that similar late positivities with a (left) frontal scalp distribution have been observed when words are not predicted but semantically possible, given the preceding context (*Federmeier et al., 2007*; *DeLong et al., 2011*; *Thornhill & Van Petten, 2012*; *Van Petten & Luka, 2012*; *DeLong, Quante & Kutas, 2014*). As noted by Van Petten and Luka, and as is the case in our data, frontal late positivities follow an N400—which is not always the case for posterior positivities. This co-occurrence is taken as an index for the sensitivity of frontal positivities to semantic predictability.

The exact functional significance of anterior late positivity is still under debate. Most researchers agree that it signals disconfirmed lexical prediction or lexical "prediction error" (*Van Petten & Luka, 2012*; *Kuperberg, 2013*, for overviews), and that the presence of a moderately or highly constraining context that can trigger updating is a prerequisite (*Boudewyn, Long & Swaab, 2015*). Note that both requirements apply to the two conditions in which we observed late frontal positivity. Taking these constraints as given, it remains unclear what processing costs occur after disconfirmed prediction. Do they involve inhibition of the predicted word—a hypothesis formulated with quite some foresight by *Kutas (1993)*, or are processing costs due to revising and updating working memory to integrate the unexpected continuation (*Federmeier et al., 2007*; *Kuperberg, 2013*)? In an ingenious study, *Brothers, Swaab & Traxler (2015)* observed late frontal positivity for words that were not predicted by their participants—who were told to actively predict continuations of sentences and who indicated afterward whether the continuation presented was the one they predicted or not. With full, trial by trial control of prediction, Brothers et al., could distinguish between specific lexical prediction and general contextual support—which our design does not allow. Given that they also observed early (pre-N400) effects of prediction, the authors conclude that the left-lateralized anterior positivity reflects prediction-related, post lexical update and revision mechanisms. Given the importance of such mechanisms for prediction in language, such anterior late positivities should be investigated further, with better control over item characteristics as is the case in our study.

## Limitations

It is important to point out that our materials were not explicitly constructed for the distinction between ANOM-Impos and ANOM-Pos and that materials were not balanced (45 vs 105 sentence pairs). As the regression analysis showed, items differed in length, which had an impact on the late anterior positivities. As *DeLong, Quante & Kutas (2014)* sentence pairs were not rated for possibility, it is not clear whether the minor discrepancies between the results of the two studies arose from potential differences of the nouns in the ANOM conditions. We also should note that overall contextual constraint was slightly lower in our study than in DeLong et al. which we aimed to replicate. Still, despite their post hoc flavor, our results on (im)possibility provide an interesting perspective on the possibility of contextual integration of even quite implausible continuations—a good reason to consider this dimension in the future.

## CONCLUSIONS

With German materials and participants, we replicated results of *DeLong, Quante & Kutas (2014)* and showed an impact of three types of constraint in sentence processing: predictability, plausibility and possibility. We observed graded effects on the N400, with the smallest negativity for expected continuations, followed by plausible but not expected alternatives, and with the largest negativity for implausible, anomalous continuations. Next, despite both being unexpected, plausible and implausible words show different patterns of posterior late positivity, arguing for a dissociation of

predictability and plausibility. Finally, we believe that the distinction between possible and impossible continuations, both being implausible, should be taken into account in studies on prediction and processing words in context.

## ACKNOWLEDGEMENTS

We are deeply grateful to Katherine DeLong and Marta Kutas for their invaluable support, and thank Dan Ke, Christian Bürger, René Michel and Daniel Kluger for their assistance in data collection and analysis.

### Funding
The authors received no funding for this work.

### Competing Interests
The authors declare that they have no competing interests.

### Author Contributions
- Laura Quante conceived and designed the experiments, performed the experiments, analyzed the data, contributed reagents/materials/analysis tools, prepared figures and/or tables, authored or reviewed drafts of the paper, approved the final draft.
- Jens Bölte conceived and designed the experiments, analyzed the data, contributed reagents/materials/analysis tools, approved the final draft.
- Pienie Zwitserlood conceived and designed the experiments, authored or reviewed drafts of the paper, approved the final draft.

### Human Ethics
The following information was supplied relating to ethical approvals (i.e., approving body and any reference numbers):

The study protocol was conducted in accordance with ethical standards of the Declaration of Helsinki and approved by the local ethics committee of the University of Münster (approval number #2016-42-LQ).

### Data Availability
The raw data are provided in the Supplemental Files.

### Supplemental Information
Supplemental information for this article can be found online at http://dx.doi.org/10.7717/peerj.5717#supplemental-information.

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
