# Peer review of "Dissociating predictability, plausibility and possibility of sentence continuations in reading: evidence from late-positivity ERPs"

_PeerJ, doi:10.7717/peerj.5717_

## Round 0.1 · original submission · Major Revisions

Two expert reviewers praised the quality of the paper and the importance of the topic, but had a number of critical comments to address in a revised manuscript. Both reviewers noted that the comparison of possible versus impossible anomalous items was problematic due to various lexical differences between these two stimulus sets. Both reviewers also indicated there are limitations to how (or even if) this comparison can be interpreted. Reviewer 1 suggested running a regression model to see if the effect survives when the lexical variables are used as predictors. Reviewer 1 also noted some minor discrepancies and omissions in statistical reporting which should be corrected. Finally, I would add to the critique of Figure 3 and say the waveforms are nearly impossible to see.

Your revision should also address the other thoughtful comments and suggestions of the reviewers.

Reviewer 1 ·

Basic reporting

Meets the criteria, though see comments below about the raw data.

Experimental design

Adheres to the criteria.

Validity of the findings

Adheres to the criteria.

Additional comments

This manuscript describes a German version of DeLong, Quante, and Kutas (2014). As expected, the results replicate the earlier study, which was itself a double confirmation of a systematic literature review (Van Petten & Luka, 2012). In addition, the authors split the anomalous condition into semantically possible and impossible (uninterpretable) completions, which affected the frontal positivity to some extent.

The original study was important, so it is good to see how well the results hold up, although this study does not seem to contribute much beyond that. The manuscript is well written. My main issues are:

1. There are limits to the interpretability of the post-hoc split of the anomalous items based on semantic possibility ratings. The possible/impossible conditions were represented by different words, which may differ in other ways besides possibility (indeed, Table 3 lists confounds of word frequency, length, neighborhood size). In addition, each participant saw only 17, 16, or 12 anomalous-impossible items, maybe fewer after artifact rejection. Ideally there would be a follow-up experiment with specifically constructed (counter)balanced possible/impossible anomaly items. But perhaps the authors can run a regression model to see if the condition effect survives when the predictors word frequency, length, and neighborhood size are included as predictors? I also recommend adding effect sizes and/or confidence intervals to at least the most important comparisons.

2. The concepts of possibility and plausibility seem quite close to each other, in fact for plausibility ratings I would probably ask participants “how likely is it that the event described in the sentence would happen in real life?”. What were the examples with which participants were instructed? And how was the cutoff of 1.5 chosen for the possible/impossible distinction? How should future more controlled studies go about manipulating this dimension?

3. I also verified the data. Reading the csv file required some find and replace operations first, among other things because each line started and ended with a quote. Also, there were some incomplete condition names, and condition names that seemed to differ only by a space:
A AN ANO ANOM ANOMI ANOMP E EX EXP EXP U US USP USP
For example line 3687 says ““AN” and line 3688 say “”ANO”.

Keeping only the lines with complete condition names, I tested the possible vs. impossible comparison on the 7 left frontal electrodes and found a 0.60 microvolt difference, t(31) = 1.798, p = .082. The slight difference with the manuscript (p = .068) might be due to the samples with incomplete condition names. I recommend these be fixed for reproducibility, especially since the last letter of the condition name is critical (ANOMI/ANOMP). In addition, the basic ANOM condition would be helpful to add (this one cannot be reconstructed by averaging across ANOMI and ANOMP, because the trial numbers differed). It could also help to add an explanation of the columns and what kind of data these are (subject averages including all time samples, it turned out).

Minor comments

Discussion, line 363: “The data for the N400 from the two studies show a very similar pattern – ignoring the position of the zero line.” I agree the zero line is relatively meaningless, but it catches the reader’s attention. Would it help to leave out the bars and zero line and simply plot the data as points with error bars?

Discussion, line 394-396:
“The N400 effects show that relative to an anomalous noun, an unexpected noun that is nevertheless a perfectly plausible continuation shows a smaller negativity, because it is less of a surprise than an implausible, anomalous continuation.”
The interpretation of the N400 as reflecting surprise seems to ignore the literature from Kutas and colleagues indicating that the N400 is not an anomaly detector (including the seminal review cited a few lines later).

Figure 3 has a lot of white space. I would increase the size of the waveforms so the data are easier to inspect.

Figure 7 is not very insightful, because the error variance depicted is between-subjects while the design is within-subjects. If the authors want to show the relevant variability they could for instance plot the variation in difference scores between conditions instead.

Reviewer 2 ·

Basic reporting

Yes - for the most part - clear, unambiguous, professional English language was used. Exceptions are noted in the comments section.
Yes, the introduction and background section do a good job at providing context for the study. The literature was well referenced and relevant.
Yes, the structure conforms to the discipline norm.
Yes, the figures are relevant, high quality, well labelled and described.
Yes, the raw data was supplied. n.b. They supplied ERP data rather than the single trial EEG data, but I thought that this was fine.

Experimental design

Yes, the paper describes original research within the scope of the journal.
Yes, the research question was well defined, relevant, and meaningful. The authors state how the research fills an identified knowledge gap.
Yes, the research described in the manuscript involved a rigorous investigation performed to a high technical and ethical standard.
Yes, the methods were described with sufficient detail and information to replicate.

Validity of the findings

The findings provide meaningful replication, as noted below.
Yes, the data is robust, statistically sound, and controlled.
In general, the conclusions are well stated, linked to the original research question and limited to supporting results. The one exception is noted below.
Speculative conclusions did not feature prominently in this manuscript.

Additional comments

The present study involves a conceptual replication and mild extension of earlier work by DeLong, Quante, and Kutas (2014) examining ERPs to words in sentence contexts that varied in their predictability and their plausibility. The goal of the study was to examine claims regarding the functional significance of two putative ERP components elicited by words in sentences: a late frontal positivity hypothesized to be elicited by unexpected but plausible continuations of a sentence, and a late posterior positivity hypothesized to be sensitive to the plausibility of the continuation. There is no firm consensus on the functional significance of such late positivities in the ERP language literature and consequently the present study — while rather unoriginal — really does address a timely issue. Moreover, the authors have done two things to extend the earlier work from the Kutas lab. First, they have constructed materials in German modeled on the earlier work, and tested healthy adult German speakers. Second, they have conducted an additional analysis of their anomalous continuation condition, comparing ERPs to continuations that are implausible but possible with those that are implausible and impossible.

EEG was recorded from the scalp as healthy adults read short texts. ERPs were formed to critical words in sentences that comprised 1) expected continuations, 2) unexpected plausible continuations, or 3) unexpected implausible continuations. The latter category was divided into roughly 50 impossible continuations, and approximately 100 continuations that were simply implausible. During the N400 interval, 250-500ms post-stimulus, unexpected plausible continuations elicited more negative ERPs than expected ones; from 550-1000ms post-stimulus, unexpected plausible continuations elicited more positive ERPs over anterior left sites. Unexpected implausible continuations were more negative than unexpected plausible ones 300-450ms; unexpected implausible continuations were more positive than unexpected plausible items over posterior scalp sites 600-1050ms, but less positive than unexpected plausible continuations over anterior lateral sites 700-900ms.

Comparison of the two subsets of unexpected implausible stimuli (viz. implausible/possible versus impossible) revealed no differences during the N400 interval, and no differences over posterior sites 600-1000ms post-stimulus. Differences over anterior left sites 600-1000ms approached but did not reach significance. Other comparisons suggested that the positivity 600-1000ms over left anterior sites was similar in amplitude for unexpected plausible items and for the possible subset of implausible items, whereas the impossible implausible items resembled the expected items. However, as far as I could tell, the stats did not unequivocally support this characterization. If this is not the case, the authors should change the manuscript to better support their case.

The materials in this study have been constructed very carefully, and the quality of the ERP data is excellent. The analysis was quite careful, and I like the way the authors combine mass univariate tests with analysis of variance. The replication of the original DeLong et al. (2014) study with German materials and speakers is valuable for the reasons noted above. The dissociation between the two late positivities (frontal versus posterior) and their association with (respectively) unexpected plausible versus implausible sentence continuations is a valuable contribution to this literature.

By contrast, the novel contribution of the present study — viz., the comparison of possible versus impossible anomalous items — is its weakest component. There were nearly twice as many possible as impossible items, and the two groups of stimuli differed in word length, word frequency, and number of orthographic neighbors. These subtle confounds and the ambiguity of the analyses of the amplitude of the late frontal positivity limit the conclusions that can be drawn about differences in the way that language users process possible versus impossible events. The conclusion in line 500 of the paper is thus not fully supported by the data. Likewise, some of the discussion in section 4.4 should be toned down.

Overall, this manuscript is impressive for the quality of the data as well as the quality of the writing. My main issue is the treatment of the two kinds of anomalous continuations as being definitively different, when the data do not fully support this characterization. Minor revision to address this shortcoming would improve the manuscript.

Lines 62-63 are a bit awkward. Perhaps put “the EEG signal should reflect” right after the comma on p. 62 and revise accordingly.
The sentence in lines 64-66 is similarly awkward. It might help to start the sentence with “Van Petten and Luka” instead of with “Post-hoc assessment”.
Lines 128-129: It’s not clear what “siding up” means in this context.
Line 335: AMON should be ANOM
Line 411: shouldn’t ‘negativities’ be ‘positivities’?
Line 474: one of the ‘both’s should be eliminated (confusing because there are three arguments)
Line 497: ‘being both’ should be ‘both being’

---

## Round 0.2 · Minor Revisions

Reviewer 1 was satisfied with your revised manuscript, but still had some questions about the labeling of condition names in one of the shared data files, and a very minor issue with a t-test value.

Reviewer 1 ·

Basic reporting

Meets the criteria, but see comments about the raw data.

Experimental design

Meets the criteria

Validity of the findings

Meets the criteria

Additional comments

The authors have been responsive in dealing with most of my comments. However, one of the shared data files still has the incomplete condition names. As mentioned before, missing letters in the condition names are an issue because the final letter distinguishes between the critical ANOMI and ANOMP conditions. Also, from the newly added file I could not reproduce the results exactly.

1) In the file with supposedly 4 conditions, here are the condition names and how often they occur (note the first line is a space only):

2
A 9
AN 32
ANO 78
ANOM 146
ANOMI 16252
ANOMP 16249
E 6
EX 14
EXP 50
EXP 16314
U 5
US 20
USP 55
USP 16304

Not surprisingly, the stats for ANOMI vs. ANOMP excluding the incomplete labels, t(31) = -1.7979, p = .08193, are still different from those in the manuscript, which says “(t(31) = 1.89, p = .068)” on p. 17. It seems this is the same result I got in my previous review.

2) The file with 3 conditions seems fine in terms of condition names:
ANOM 16384
EXP 16384
USP 16384

However, when I test “EXP “ against “USP “ as described in the manuscript, the t value I get is t(31) = -3.6639. This differs from that reported in the manuscript, which says “Paired t-tests revealed that EXP nouns differed from USP nouns (t(31) = -3.97, p < .001)” (p. 16). Perhaps this has something to do with rounding errors or with including/excluding the 1000 ms sample at the edge of the time window (I included it), but the authors will want to double-check their analyses and make sure they align with (and can be reproduced using) the shared data file.

---

## Round 0.3 · accepted · Accept

Thanks for your rapid response.

#